# A New Application of Social Impact in Social Media for Overcoming Fake News in Health

**DOI:** 10.3390/ijerph17072430

**Published:** 2020-04-03

**Authors:** Cristina M. Pulido, Laura Ruiz-Eugenio, Gisela Redondo-Sama, Beatriz Villarejo-Carballido

**Affiliations:** 1Department of Journalism and Communication Studies, Universitat Autònoma de Barcelona, Campus de la UAB, Plaça Cívica, 08193 Bellaterra, Barcelona, Spain; cristina.pulido@uab.cat; 2Department of Theory and History of Education, University of Barcelona, Gran Via de les Corts Catalanes, 585, 08007 Barcelona, Spain; 3Faculty of Social and Human Sciences, University of Deusto, Unibertsitate Etorb., 24, 48007 Bilbo, Bizkaia, Spain; gisela.redondo@deusto.es; 4Faculty of Psychology and Education, University of Deusto, Unibertsitate Etorb., 24, 48007 Bilbo, Bizkaia, Spain; beatriz.villarejo@deusto.es

**Keywords:** social media analysis, social impact in social media, public health, fake news, vaccines, nutrition, Ebola

## Abstract

One of the challenges today is to face fake news (false information) in health due to its potential impact on people’s lives. This article contributes to a new application of social impact in social media (SISM) methodology. This study focuses on the social impact of the research to identify what type of health information is false and what type of information is evidence of the social impact shared in social media. The analysis of social media includes Reddit, Facebook, and Twitter. This analysis contributes to identifying how interactions in these forms of social media depend on the type of information shared. The results indicate that messages focused on fake health information are mostly aggressive, those based on evidence of social impact are respectful and transformative, and finally, deliberation contexts promoted in social media overcome false information about health. These results contribute to advancing knowledge in overcoming fake health-related news shared in social media.

## 1. Introduction

Fake news has been defined as fabricated information that imitates news media content in form but not in organizational process or intent, which overlaps with other information disorders, such as misinformation—false or misleading information—and disinformation, which is false information that is deliberately disseminated to deceive people [1]. The impact of fake news in social media is a major concern in public health, as it can reduce or increase the effectiveness of programs, campaigns and initiatives aimed at citizens’ health, awareness and well-being. The advancements in the methodologies related to social media analysis provide new insights to unveil how citizens share health information and the ways in which fake news influences public health. Social impact in social media (SISM) constitutes a novel methodology in both social media analytics and the evaluation of the social impact of research [2]. This article applies the SISM methodology to the specific case of fake news in health to identify the type of interactions related to the information shared in social media.

### 1.1. Impact of Fake News on Public Health

Fake news concerning health on social media represents a risk to global health. The WHO warned in February 2020 that the COVID-19 outbreak had been accompanied by a massive ‘infodemic’, or an overabundance of information—some of which was accurate and some of which was not—which made it difficult for people to find reliable sources and trustworthy information when they needed it [3]. The consequences of disinformation overload are the spread of uncertainty, fear, anxiety and racism on a scale not seen in previous epidemics, such as SARS, MERS and Zika. Therefore, the WHO is dedicating tremendous efforts aimed at providing evidence-based information and advice to the population through its social media channels, such as Weibo, Twitter, Facebook, Instagram, LinkedIn and Pinterest, as well as through its website. The MIT Technology Review highlights that social media are not only being used to spread false news and hate messages but are also being used to share important truthful data and solidarity with all those affected by the virus and hate messages [4].

We are in what some have called the second information revolution [5]. The first information revolution began with the spread of the written word through the press. Now, in this second information revolution, a digital transformation is shaping how citizens around the world interact with each other. We are facing an unprecedented global expansion in the ways we share, access and create information that is presented in many forms—one of which is social media.

From diverse fields of knowledge linked to health issues, it can be stated that social media can have both a positive and a negative impact on public health [6,7]. On the one hand, the combination of artificial intelligence and big data can help public health providers identify pandemic diseases in real time, improving the coordination of the response of public health systems through information sharing and improving surveillance and diagnostics. Furthermore, research shows that social media information and social media responses are effective strategies to gain feedback on potential public health policy proposals. This positive impact of social media in health has been demonstrated in a recent study about heat protection policy for Australian schools, which, through the analysis of public comments posted on a national Australian Broadcasting Corporation (ABC) website, identifies the themes to support a national heat protection policy for schools [8].

There is evidence of the negative effects of using social media to spread misinformation, which produces harmful consequences to global health and well-being, becoming one of the greatest challenges for public health systems today [5,9]. The most extensively studied topics involving misinformation in health are vaccination, Ebola and the Zika virus, as well as nutrition, cancer, the fluoridation of water and smoking [5,10,11,12]. Disinformation spread by the anti-vaccine movement has led to episodes regarding vaccination provoking easily preventable disasters, such as the measles epidemic in Washington state (January 2019). The spread of false information also explains a decrease in immunization behaviors with respect to measles-mumps-rubella (MMR) vaccinations, explaining the drop in the demand for this vaccine in the UK and the USA between 1999 and 2000 [13].

Furthermore, research related to the negative impacts linked to the authenticity of social media and identities has increased in recent years [14]. This includes the analysis of the problems surrounding social media messages/posts regarding privacy, posts ending with unintended users, concerns on how to use social media platforms, who to follow and how people portray themselves in an inauthentic manner [14,15].

### 1.2. Spreading Fake News on Health in Social Media

#### 1.2.1. The Context in Which Fake News Is Spread

Never before in human history has the role of globalization processes had the impact that it currently has in decision-making processes and societies because of the speed of communication [5]. Globalization also plays a crucial role in the spread of health news, including social media, influencing the way users receive such news [16,17]. In this arena, it is important to highlight that in a globalized world, health content information can be perceived differently depending on the target group or context [18]. Therefore, fake news may reach citizens in different ways, depending on their age, culture, and other factors [19]. Moreover, research shows that social media and related global digital media content influence discourses about professions and how citizens perceive them, including public health professionals [20]. For instance, many public health programs aimed at children and youth have physical education at the core of their initiatives [21,22]. The teaching profession is often portrayed in digital media in relation to unhelpful physical crisis messages [23] or discourses related to bullying in movie scenes. In a similar vein, social media has also been demonstrated to influence the perceptions of adolescent students with regard to their sexual and reproductive health learning [24]. As a result, health professionals may recognize that social media channels, such as Facebook, offer possibilities to support their activities.

Research on the role of mass media and messages and dominant discourses that are communicated to the public is an emerging topic of interest in scientific works that requires further investigation. The influence of social media discourses may differ depending on age, culture or gender [19]. For example, young people build their identities, construct knowledge and acquire information from digital media, including social media, beyond formal education and classroom learning, which is an approach resonating with “public pedagogies” [25]. Other authors such as Ulmer [26] argue that mass digital media provides the public an ‘entry point’ into the debates. The fact is that digital and social media contribute to the strength or undermine the diversity of points of view, influencing the development of specific health public health policies and interventions [27,28]. Such influence of the media has been defined by some authors as the “fourth state” [29]. Taking into account these contextual considerations, it is important to face fake news related to health in social media to support public health policies instead of trying to reverse them.

#### 1.2.2. Fake News, Health and Social Media

In a globalized world, the spread of fake news content on health-related topics in social media and the ways in which it spreads have recently been discussed in depth [10,30]. Misinformation and disinformation—misinformation as inaccuracy and errors and disinformation as a falsehood created on purpose and the spread of it by malicious individuals (human or bots)—gain momentum from the desire to find a solution to a particular disease or illness by patients or their relatives, who inadvertently contribute to spreading misleading information.

Globally, the narratives of misinformation are dominated by personal, negative, and opinionated tones, which often induce fear, anxiety, and distrust of institutions. Once misinformation gains acceptance in such circumstances, it is difficult to correct, and the effectiveness of interventions varies according to the personal involvement of each individual and his or her literacy and sociodemographic characteristics [10]. However, other studies have shown that ignorance rarely leads to strong support for a cause. For example, those who most strongly reject the scientific evidence of climate change are also those who believe that they are best informed about the issue. People’s pre-existing attitudes often determine their level of belief in misinformation [31].

With respect to globalization processes, evidence suggests that false information spreads globally more pervasively and farther and faster than the truth spreads in social media. In examinations of possible explanations for this global phenomenon, it has been found that novelty is a pervasive component of false rumors, which are significantly more novel than the truth. However, data cannot support the contention that novelty is the only reason, or the main reason, for the spreading of falsehood [13,32]. However, other studies that have focused on the analysis of fake news in social media have reached different conclusions [33,34].

A recent study that analyzed the credibility of sources publishing articles online that may reach global audiences concludes that for the specific case of online health information and content on social media, people are more concerned about the veracity and credibility of the information source and tend to spread less misinformation about health. One possible explanation given by the authors is that people generally do not read health information for entertainment but rather search for information useful to their health or that of people close to them. Furthermore, in these cases, they are less likely to have a pre-existing opinion about health information than are those who share fake news stories about other topics [34]. A similar conclusion came from a fact-checking study of Twitter and Sina Weibo (the most-used social media platform in China), developed 24 hours after the WHO’s declaration of the Ebola outbreak as a Public Health Emergency of International Concern in August 2014.

In a globalized world, this declaration by the WHO had diverse impacts on the definition of private and public strategies to combat the virus. It contended that only 2% of the posts created on Twitter and Weibo were fake news or disinformation, while the rest were outbreak-related news and scientific health information, mostly coming from news agencies reporting information from public health agencies. This study was able to confirm that these two social media sources contributed to spreading the news of the Ebola outbreak, which was the key message of the WHO [33].

Research on fake news on health in social media covers a variety of channels, including Twitter, Facebook, Reddit, and Weibo [35,36,37]. The analysis of Twitter has gained special attention, and research shows the reach of Twitter codes and the manner in which information spreads on Twitter. This occurs in diverse fields related to public health, from physical education [38] to healthy eating habits or healthy lifestyles [39].

### 1.3. Identification of Social Media Interactions as Key to Spreading or Combating Health-Related Fake News

Social interaction appears to be the main method of understanding how disinformation or fake news spreads over social media. Different studies have been conducted to identify by who and how health disinformation content is promoted in social media. In the case of Twitter, different types of malicious actors covering both automated accounts (including traditional spambots, social spambots, content polluters, and fake followers) and human users, mainly trolls, have been identified. It is very difficult to detect whether there is a human or a bot behind a profile. However, all of them produce distorting effects that may be critical to messages from public health systems [12].

One of the studies in the case of vaccines identified three types of profiles that had a special probability of spreading vaccine-related disinformation [12]. The first profile is trolls, or Twitter accounts with real people behind them, identified from lists compiled by U.S. authorities; these trolls use the hashtag #VaccinateUS and spread pro- and anti-vaccination messages, often with the apparent aim of encouraging people to believe that the medical community is divided. The second profile, called ‘sophisticated bots’, is artificial intelligence that automatically spreads content via Twitter with the same objective of making people believe that the medical community is divided. The third group of profiles is comprised of content polluters, who use anti-vaccine messages to pique users’ curiosity and lead them to click on links, such that every click leads to more income for those behind the website. Some studies have suggested the need to increase social media literacy, provide strategies and instruments to check the reputation, consistency, and evidence of any information, and avoid self-confirmation (based on assumptions or previous unchecked experiences) [40,41].

### 1.4. Combating Fake News on Social Media

Several approaches have been proposed in recent years to automatically assess credibility in social media. Most of them are based on data-based models, i.e., they use automatic learning techniques to identify misinformation. Based on these techniques, different applications have been developed with different objectives and in different contexts, such as detecting opinion spam on review sites, detecting false news and spam in microblogging, and assessing the credibility of online health information [6,42,43]. These techniques include both human intervention and algorithms to verify the veracity of information across technologies, such as artificial intelligence (AI) and natural language processing (NLP) [44]. Other mechanisms developed as a remedy against fake news on social media are source ratings that can be applied to articles when they are initially published, such as expert ratings (in which expert reviewers fact check articles—the results of which are aggregated to provide a source rating), user article ratings (in which users rate articles—the results of which are aggregated to provide a source rating), and user source ratings (in which users rate the sources themselves) [34].

According to the literature, social media is an interaction context in which misinformation is spread faster, but at the same time, there are interactions focused on health that are evidence based. Furthermore, it is important to highlight that social media users share the social impact of health research. However, less is known about what these interactions are or what type of recommendations we can identify from them.

The main aim of this work is to contribute to advancing methods of overcoming misinformation or fake news in health through social media data analysis. In this sense, we have applied the methodology of social impact in social media (SISM) [2] to identify which type of interactions spread misinformation and which type of interactions contribute to overcoming fake news or misinformation related to health. The main research questions are as follows: RQ1) How are social media messages focused on fake health information or misinformation?; RQ2) How are social media messages focused on health evidence with potential or real social impacts?; and RQ3) How do interactions based on health evidence with potential or real social impacts help overcome misinformation or fake health information? The results indicate that messages focused on fake health information are mostly aggressive, those based on evidence of social impact are respectful and transformative, and finally, deliberation contexts promoted in social media overcome false health-related information.

## 2. Materials and Methods

The method used and the sample selection are described in detail in the following sections. The method used is social impact in social media (SISM) methodology [2], which combines quantitative and qualitative content analysis of the sample selected considering the contributions of the social impact of the research [45]. According to Elo et al. [46], there are trustworthiness issues in the preparation phases of the data collection method, the sampling strategy and the selection of a suitable unit of analysis according to the way in which the research goals are defined. In this sense, the following sections develop in detail how the sample was selected and how the data collection and extraction were developed. The data analysis is explained in detail in the corresponding section, along with an explanation of the unit of the analysis used.

Regarding ethical considerations, the present research adheres to international ethical criteria related to social media data collection and corresponding analyses; in particular, we have followed the ethical guidelines for social media research supported by the Economic and Social Research Council (UK) [47] and Ethics in Social and Humanities Sciences of the European Commission [48]. Furthermore, we have perceived the risk of harm to and conserved the anonymity of users. Additionally, we have read the terms, conditions and legalities of each of the social media channels, and we have used only public information without identifying any user.

Likewise, the data were appropriately coded and anonymized to avoid the possibility of traceability. Sets of data have been secured, saved, and stored. The dataset analyzed and the calculations performed are available in the Appendix A (dataset). We cannot share all raw data due to the current terms of the social media channels and the General Data Protection Regulation (GDPR).

### 2.1. Sampling, Social Media Data Collection and Extraction

The first step to develop this study was the selection of a suitable sample of social media channels to collect the data. The social media channels for the analysis are Facebook, Twitter and Reddit, and their selection corresponds to three criteria: (1) relevance of the number of active users in millions according to Statista 2019 data: Facebook (2414), Twitter (330), Reddit (330); (2) availability of public messages; and (3) suitability for online discussion. There are other social platforms with millions of users, but these three have been selected because they are more suitable for our research study.

The chosen sample is exploratory and selective and is based on the following criteria.

Criteria 1: Selection of suitable searchable keywords. We selected the word “health” as a general topic and the specific keywords “vaccines”, “nutrition” and “Ebola”. The use of these specific keywords is based on the findings by Wang et al. [10], in which the authors identify vaccines, nutrition, and Ebola as topics with more misinformation in social media. Specifically, we used the hashtags #vaccines, #nutrition and #Ebola to extract Twitter information. In relation to Facebook, we selected two public pages with more audiences in relation to the topic “health”. With regard to Reddit, we selected the topic “vaccines” in a community focused on this topic and an “AskScience Ama Series” focused on vaccines.

Criteria 2: Data extraction. The data extracted from Twitter contain tweets published under the hashtags selected in the last ten days. In the case of the Facebook page, the data are extracted from the last 100 posts published and the corresponding comments of the two Facebook pages selected. Finally, in Reddit, we selected the comments published in two conversations in two different communities (the AskScience Ama Series focused on vaccines and the vaccines subreddit). Table 1 shows the data collected.

### 2.2. Data Analysis

The strategy for data analysis aims to unveil the nature of interactions focused on misinformation or fake health information and the nature of interactions based on health evidence of potential or real social impacts. To do so, we have designed the following steps and strategy. The unit of the analysis is the full message published by the user (tweets, Facebook posts and Reddit messages (posts and comments)), which means that the information provided in the external links included in the messages is also analyzed.

#### 2.2.1. Steps

Step 1: To identify which tweets (top 100 posts for each hashtag) and Facebook posts (top 20 posts) have received more attention. In the case of Twitter, this identification depends on likes and retweets. In the case of Facebook, this identification depends on likes on the posts of the Facebook pages selected and the public comments with more likes (top 20). In the case of Reddit, this identification depends on the total number of interactions of 4 conversations in the vaccines subreddit and the 100 most-valued comments by the community (the comments are sorted by reader preference), and each comment receives points given by different members of the community on the AskScience Ama Series. For the case of the AskScience Ama Series of one of the conversations, we selected the last 20 comments sorted by reader preference, such that each comment receives points according to the preferences of different members of the Reddit community. Table 2 shows the data selected.

Step 2: Development of qualitative content analysis for each message selected (N = 453) (tweets, Facebook posts, Facebook and Reddit comments). Researchers apply a classification of messages according to the codebook (see Table 3) and interactions received. The social impact coverage ratio (SICOR) will be applied for each source of social media selected, which identifies “the percentage of tweets and Facebook posts providing information about potential or actual social impact in relation to the total amount of social media data found” [2]. The elaborated codebook has four categories defined a priori as a result of the literature review performed. The categories classify messages analyzed regarding evidence of social impact, false news, misinformation, opinion, and facts. While the research team performed the first analysis with these four categories, two new categories emerged from the analysis. These two categories were messages that ask for evidence of social impact and messages that contain misinformation but search for dialogue to contrast with the information—both messages search for deliberation.

Step 3. In-depth analysis of interactions containing evidence of potential or real social impact.

#### 2.2.2. Interrater Reliability (Kappa)

The analysis of social media data collected for the second analysis was conducted following a qualitative content analysis method, in which reliability was based on a peer-reviewed process. The sample was composed of 453 messages. Each message was analyzed to identify whether it contains evidence of potential or real social impact (ESISM = 1), if it was a message of misinformation or fake information in health (MISFA = 3), if it was an opinion (OPINION = 4) or if it was information (INFO = 2). The researchers involved were experts in the social impact of research and fake news. Each researcher was provided with the codebook before starting to code the messages. Once the analysis was finalized, the messages were coded and compared. We used interrater reliability in examining the agreement between the two raters on the assignment of the categories defined using Cohen’s kappa. The result obtained was 0.79; considering the interpretation of this number, our level of agreement was almost perfect and, thus, our analysis was reliable. In cases where no agreement was achieved, the raters decided to exclude the results (N = 450).

## 3. Results

Before answering the research questions, there are initial steps to determine whether, among the messages selected in the sample, there are more messages and interactions based on misinformation or, on the contrary, there is more evidence of potential or real social impact. For this purpose, we first classified the messages (see Table 4), and second, we calculated the SICOR.

In relation to the Twitter sample analyzed, we found that tweets with a higher percentage of ESISM were those that were published under hashtag #Ebola (39%), followed by #nutrition (18%). In the case of #vaccines, a lower percentage of ESISM was found (only 4%), but MISFA had a higher percentage (32%). The higher percentage of three hashtags selected is the node of INFO, and OPINION is higher in the #vaccines hashtag with 29%.

In relation to the data analyzed on Facebook, the lower percentage is under the code of MISFA (5% on Facebook page 1 and 3% on Facebook page 2), and the percentage of INFO is higher on Facebook page 1 than on Facebook page 2. However, OPINION is higher on Facebook page 2. The case of ESISM is only present in the case of Facebook page 1 with 15%.

In the case of subreddits, we selected examples focused on vaccines because this topic was the most controversial on Twitter. One of the results indicates that the percentage of ESISM (27% and 14%) was higher than the MISFA code (6% and 10%), INFO had the highest percentages in the subreddits (37% and 62%), and OPINION had the second-highest percentages (31% and 14%).

If we analyzed the total amount of data collected, we obtained the following SICOR for each social media channel selected, as shown in Table 5. The SICOR calculation [2] is a ratio that calculates the percentage of ESISM found in the full sample selected. In this case, the SICOR is the percentage of tweets with evidence of social impact in relation to all the tweets collected; the same applied to Facebook posts and comments and subreddit comments.

As we can see, the selected subreddit comments have a higher percentage of SICOR (23%), followed by tweets (20%) and Facebook posts and comments (8%).

In the case of MISFA, the percentage of total amount for each social media channel selected is shown in Table 6.

According to the results, the MISFA percentage is higher in tweets (19%), followed by subreddit comments (7%) and Facebook posts and comments (4%).

### 3.1. Fake Health Information Social Media Messages are Mostly Aggressive

Regarding the research questions, the nature of the social media messages focused on false health information is that they are mostly aggressive in the sample analyzed. This result is mainly concerning the messages on vaccines in which the possibility of dialogue does not exist; there is no option, and the messages contained an affirmative closed sentence. First, we have detected hostility to arguments based on science and even defamation of scientists who have contributed to advances in the field of vaccines. One of the examples of fake news in vaccines said the following: “Vaccines have a long history of damaging the brain from day one”, followed by “The-so-called Father of Vaccination left his first son brain-damaged by vaccinating him, Jenner was smart enough not to vaccinate his second”. This is an example false of information that has negative impacts on the truth. First, Jenner devoted his life to overcoming smallpox. Jenner even freely treated poor people to save them from smallpox. His discovery had a substantial social impact, and his sons died due to tuberculosis, not due to smallpox vaccines. This scientific article explains in detail the contribution of Edward Jenner [49] and concludes how his discovery and the promotion of vaccination facilitated the eradication of smallpox. Thus, spreading false information about this crucial discovery with defamation of real history harms citizens’ lives because this lie could damage their health.

Another example found that parents should be encouraged to boycott doctors who recommend vaccinating their children. There is an active anti-vaccine movement that is continually sharing this type of message in social media. The negative consequences are that some children who are not vaccinated have contracted diseases that could be avoided, in addition to adults who have done the same. For instance, in 2019, there were 1282 cases of measles in an outbreak in 31 states [50]. This number was the highest number of cases reported in the U.S. since 1992. Most cases occurred among those who were not vaccinated against measles [50]. The negative impact of this type of interaction affects the public health of cities and villages where people decide to follow these anti-scientific arguments.

For this reason, these types of messages are also aggressive, as they do not to follow scientific arguments related to health and cause physical damage and disease among children and those who share a common space. Regarding the examples of the type of MISFA, we found some messages that contain false information or misinformation but contain questions to open dialogue that may pose contrasts to one’s own assumptions with scientific evidence delivered by other persons or the beginning of deliberation. For instance, one of the examples begins with “I would be interested in a healthy and respectful conversation”, saying that he is a “vaccine agnostic”, sharing his opinion that he believes there are more risks than benefits. However, he is open to dialogue. This message is based on opinion, not scientific evidence. However, he is honest, saying that it is an opinion, and he is not assuming that he knows the truth; this is a first step towards dialogue. This case reflects another type of message found, people who are influenced by false information but open to having a conversation. Nevertheless, this opinion also has negative consequences, as he stated that his children are not vaccinated and, thus, they are also at risk.

### 3.2. Potential or Real Social Impact Social Media Messages Are Respectful and Transformative

Messages that contain the potential or real social impact of health are respectful and transformative. They deliver quantitative or qualitative evidence of the social impact that contributes to knowing health is being improved. Some of the illustrative examples of this are those published under the Ebola topic. One of the examples analyzed was the impact that the Ebola vaccine was finally approved. Ebola is a health concern, especially in DRC, due to the number of people affected and who die due to this disease. One of the examples shares quantitative evidence of the social impact of this vaccine and congratulates the people that made this result possible “Merck’s Ebola vaccine, which has been given to more than 258,000 people in the current outbreak in DRC”.

Another example of this type of message said that the Ebola vaccine is the best of 2019. This discovery offers hope and optimism for overcoming Ebola in DRC by offering qualitative evidence of the potential social impact of this vaccine. For example, a survivor of Ebola who took part in the vaccine trial was quoted as saying, “I can convince other people in my town that there is a treatment available for Ebola and that they can get better”, and a link was added to the WHO article with the full testimony of this survivor [51]. Similarly, another example of qualitative evidence of social impact is delivered by a message that contains a documentary of Ebola through different testimonies, such as Jophet Kasere, who survived Ebola. However, his family did not; he works as a nurse, caring for children whose parents have been infected with the virus. The documentary recorded by Frontline shows how treatment delivered by WHO was improving the health of different members of the community, but at the same, shows how people who were against this international help tried to stop this improvement and leave people at risk [52].

### 3.3. In Deliberation Contexts, Messages with Evidence of Social Impact Overcome Fake Information in Health

One of the results found is that deliberation contexts in social media promote the possibility of contrasting information and open dialogue based on valid claims. This example has specially been observed in the Reddit conversations analyzed. The social network allows conversations abiding by the rules of the communities. For instance, one of these rules states that conversations should be based on scientific information and not on false information. We have found examples of people with doubts or concerns regarding vaccines using Reddit to share their views and learn. In Reddit, we found MISFA D and ESISM D, because the common goal is to dialogue.

For instance, one of the examples found was a conversation initiated by a girl who was not vaccinated. She said “my parents never vaccinated me,” and she was concerned about this and her health. Her questions were addressed to the community, seeking help with regard to her situation. She received replies focused on helping her. For instance, she was told to visit her GP in order to receive an appropriate catch-up schedule, and the importance of talking to a doctor was stressed, “this is not something you should be deciding yourself or asking the Internet about; just ask your doctor”.

The second example selected was a conversation initiated by a person who holds anti-vaccine views but was searching for “some answers (if possible) to vaccine questions”. This person affirms that “there has never been a vaccinated Vs. completely unvaccinated study” to extract reliable conclusions about whether it is better to vaccinate or not. This person received replies with evidence of social impact focused on comparative studies between people who were vaccinated and not vaccinated, where those who were vaccinated exhibited better health than those who were not. Direct links to these studies were also provided. Some of the information detailed the following, “German study on lower rates of asthma among the vaccinated”, “comparing unvaccinated and vaccinated people who do catch the flu–vaccinated people are protected from the most serious effects, vaccinated versus unvaccinated children: how they fare in first five years of life, Nigerian study of 25 unvaccinated and 25 vaccinated children: one vaccinated child had a mild case of measles. Unvaccinated children: 3 dead, plus 11 non-fatal cases of measles”. The second reply selected detailed cases of measles in the U.S. and explained how the number of cases increased due to unvaccinated children, and this was then compared with Romania, providing scientific sources where the data are published. The result of this conversation is that the person who began this conversation read extended replies that were well-argued based on evidence of social impact and official data, replying enthusiastically, “Thanks! I’ll read these and think on the issue!” Thus a transformation was possible due to arguments based on evidence of social impact.

Another example selected is from the ASK ME SCIENCE conversations; this is a conversation where scientists are available for dialogue with citizens about different topics. In this case, vaccines were used. One of the conversations selected was concern around Andy Wakefield and his research—that is, an ex-physician who became an anti-vaccine activist, among those responsible for purporting a link between vaccines and autism. The clear and overwhelming consensus among scientists is that “His malevolent influence on the vaccine world was terrible, and we have still not fully recovered even though his publications and ethics have been debunked. Because of his paper, millions of people were not vaccinated, and thousands have died. What a legacy to live with”.

Furthermore, the final example selected was regarding the negative impact of a community that opts to remain vaccinated. This dialogue was started by someone sharing the concept of “herd immunity”. This concept details that in a community, there are people who cannot be vaccinated, such as due to allergies or those who are immunocompromised, and they depend on herd immunity to protect them. This was followed by highlighting that “If too many people who could get the vaccine but choose not to, it does not just affect the individual but can compromise others in the community as well”. This person explained that in his/her county, a large population chose not to vaccinate and, consequently, there was a measles outbreak; further, an emergency was declared. He/she has a friend who is immunocompromised and needed to stay home for fear of contracting the measles, “it was terrifying and preventable if people who could get the vaccine would choose to do so. It is a choice that does have an impact on others”. This conversation opens a dialogue on how our decisions based on false information can have a negative impact on the health of others. Thus, it is crucial to apply the evidence of social impact in collective matters to guarantee, in this case, successful public health.

## 4. Discussion

The previous studies reviewed have been useful in clarifying how health information is spread in social media, identifying the positive and negative impacts [6,7]. Regarding the adverse effects of using social media to spread misinformation, there is evidence of the harmful consequences to global health and well-being, becoming one of the most significant challenges for public health systems today [5,9]. Some of the studies have advanced the identification of the types of profiles that spread vaccine-related disinformation [12], and this helps to identify whether the profile that is posting could be a trusted source or not. Our study contributes to advances in the direction of overcoming false information in health through the analysis of how the messages and interactions are based on false health-related information and the transformative dimension of those messages based on evidence of social impact. This identification has made it possible to apply the SISM methodology, which is focused on evidence of social impact. The three social media channels show that there is a public online discussion regarding the object of study. The detailed analysis of the selected sample allowed us to identify deliberation contexts in the three social media channels; for instance, in Reddit, the open conversations encourage people to search for a dialogue based on valid claims.

Moreover, in this context, messages based on evidence of social impact overcome false information, even among those with previous anti-vaccine ideas but with an open-minded attitude and respect. However, it is not possible to engage in dialogue with those who have an aggressive position against science. This finding is especially crucial because it allows us to identify whether citizens have access to evidence on social impacts and whether they can share this evidence in conversations in which false information is spread. The evidence of social impact is the vaccine against false health-related information. Future research lines could replicate this analysis in other topics in which false information is damaging. Nevertheless, civil rights movements could also promote these findings to quickly overcome false health-related information that is causing deaths in adverse but avoidable situations.

On the basis of the research findings, there are several practical implications and recommendations for public health professionals. First, the results allow public health professionals to determine the type of health information with evidence of social impact that is most shared in social media. Second, the results also contribute to understanding the types of fake news with a stronger presence in social media that can reduce the effectiveness of public health social media campaigns. Third, this knowledge can be useful in the design of strategies in the public health sector to reverse fake news. Fourth, this knowledge can also be useful to narrow efforts to disseminate evidence of social impact in health to deactivate fake news. Finally, this study contributes to identifying discussion forums in which debates are occurring around health information to contribute to the dialogue providing health information with evidence of social impact.

## 5. Conclusions

This article demonstrates that SISM is a replicable methodology that has been successfully applied in social media analytics in relation to health and fake news, contributing to the further exploration of the possibilities of this methodology. This study offers the possibility to identify, on the one hand, evidence of social impact shared in social media and, on the other hand, misinformation or fake information related to health. Furthermore, the results show how the interactions in social media depend on the type of information shared or commented upon by diverse actors.

The analysis of Twitter, Facebook and Reddit unveils the different types of interactions regarding evidence or fake news, but they all have the common pattern of showing more messages of events or fact-related information about Ebola, nutrition and vaccines. Furthermore, in most cases, the existence of interactions regarding evidence is higher than that of interactions regarding the misinformation of fake information, although the percentage is much higher for the misinformation of fake information than for evidence in the case of Twitter #vaccines. With regard to opinions, the results indicate that they are much more frequent on Facebook and on subreddits than on Twitter. Moreover, the percentage of tweets and Facebook posts providing information about potential or actual social impacts in relation to the total amount of social media data (SICOR) is higher in Tweets and subreddit comments than in Facebook posts and comments. Another relevant finding is that messages focused on false information regarding health are mostly aggressive, and messages based on evidence of social impact are respectful and transformative. Finally, deliberation contexts in social media allow for the transformation of even those who have false information but who are open to dialogue when they participate and access evidence of social impact.

The findings provide insights into the way in which public health initiatives can support the presence and interactions of evidence as an effective strategy to combat fake news. Two main recommendations are suggested for public health professionals, among others. On the one hand, we narrow the dissemination strategies to reverse and deactivate fake news regarding health, considering that the percentage of misinformation on fake news is much higher than that observed for Twitter #vaccines. On the other hand, the design of concrete interventions for discussion forums in which health information is discussed (not only shared) can provide health information with evidence of social impact.

This research contributes to including citizens’ voices into research from a bottom–up approach, in line with the need to support science and social dialogue in relation to public health, including vulnerable groups [53] or the role of patients to overcome barriers to health access [54]. The possibilities of social media analysis have been widely explored in very diverse fields, from gender to digital protests [55,56], and this work contributes to advancing knowledge in social media analysis and fake news in public health. Future investigations can use SISM to analyze the interactions in social media regarding other public health issues to further explore how citizens use and share information.

## Figures and Tables

**Table 1 ijerph-17-02430-t001:** Social media data were collected.

Social Media	Keyword	Data Collected
Twitter	#vaccines	12,965 tweets
	#nutrition	200 tweets
	#Ebola	4052 tweets
Facebook	Facebook Page 1	100 posts and 10,118 corresponding comments
	Facebook Page 2	100 posts and corresponding 2958 comments
Reddit	Subreddit community	4 conversations on vaccines with 55 comments
	AskScience Ama Series	A conversation with experts on vaccines with 342 comments

**Table 2 ijerph-17-02430-t002:** Final sample selected for the social impact in social media (SISM) analysis.

Social Media	Keyword	Data Selected
Twitter	#vaccines	100 tweets
	#nutrition	100 tweets
	#Ebola	100 tweets
Facebook	Facebook Page 1	20 posts and 20 comments
	Facebook Page 2	20 posts and 20 comments
Reddit	Subreddit community	4 conversations on vaccines with 4 open comments in the last 48 comments
	AskScience Ama Series	1 Ask Ama Series with 1 open comment in the last 20 comments

**Table 3 ijerph-17-02430-t003:** Codebook.

Classification	
**Four categories defined a priori (deductive)**	
**Code**	**Definition**
ESISM	The message (tweet, Facebook posts or Reddit messages) is an example of evidence of the social impact shared in social media. This means that there is evidence of improvement in relation to the topic selected. This evidence could be a potential or real social impact of research results that is linked with societal goals, for instance, the UN Sustainable Development Goals, and that contributes to improving the specific health issue concerned. This information is useful in connecting citizens with trustworthy information. The message offers a link to evidence of contrasting information or sources of the evidence with possible contrasts.
MISFA	The message (tweet, Facebook posts or Reddit messages) is an example of misinformation or fake information in health. Both situations have negative consequences for public health and personal health. It does not offer a contribution in order to contrast but rather presents the message as evidence or trust information, or the source of the information offered is not scientific.
OPINION	The message (tweet, Facebook posts or Reddit messages) is an opinion, and the message is presented as opinion, not as evidence.
INFO	The message (tweet, Facebook posts or Reddit messages) is an event or fact, for instance, news.
**Two codes emerged from the analysis (inductive)**	
ESISM D	The message (tweet, Facebook posts or Reddit messages) is formulated as a question to ask for evidence of scientific results that ensure social impact or that are a starting point for deliberation.
MISFA D	The message (tweet, Facebook posts or Reddit messages) contains misinformation or fake information, but it contains questions that open dialogue in order to contrast one’s own assumptions with the scientific evidence delivered by other persons or to begin a deliberation.

**Table 4 ijerph-17-02430-t004:** Percentage of coded messages in relation to data collected by each social media channel selected.

Social Media	ESISM	INFO	MISFA	OPINION
Twitter #Ebola	39%	51%	3%	7%
Twitter #nutrition	18%	51%	21%	10%
Twitter #vaccines	4%	35%	32%	29%
Facebook page 1 (posts and comments)	15%	48%	5%	33%
Facebook page 2 (post and comments)	0%	43%	3%	55%
Subreddit conversations	27%	37%	6%	31%
AskScience Ama Series	14%	62%	10%	14%

**Table 5 ijerph-17-02430-t005:** SICOR.

Social Media	SICOR—Social Impact Coverage Ratio
Tweets	20%
Facebook posts and comments	8%
Subreddit comments	23%

**Table 6 ijerph-17-02430-t006:** Percentage of MISFA for each social media channel selected.

Social Media	MISFA (Misinformation and Fake Health Information)
Tweets	19%
Facebook posts and comments	4%
Subreddit comments	7%

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
