# Peer review of "A New Application of Social Impact in Social Media for Overcoming Fake News in Health"

_ijerph, 2020, doi:10.3390/ijerph17072430_

Round 1

Reviewer 1 Report

Dear authors

I thank you for a well-documented, well-written article on the impact of social media and fake news. It is a timely study that will help guide authorities to design adequate responses to "content pollution" on the Internet.

I recommend that the article is being published without further revision. 

Reviewer 2 Report

Thank you for this interesting article. It is indeed very on point with the current climate in the world.

Several full stops are missing (lines, 59 after [4,7]; line 61 after [4,8-10] and all the way throughout the paper.

Rework line 102-103 -"Different research has been conducted to identify who and how are promoting health disinformation contents in social media".

Have the article proof read for small grammatical errors - Line 324 for example, "For instance, one of the examples founded was a conversation initiated by a girl that is was not 324 vaccinated." - should be "For instance, one of the examples found was a conversation initiated by a girl that was not vaccinated". 

line 331 - initiated to one person - initiated by one person 

Reviewer 3 Report

The subject is very contemporary.
Overall, the paper is adequate and I have just few points.
I think the development of codebook used on classification process could be explained with more details.
At line 241 is missing a "%". The correct would be "27% and 14%"
The values on the second column of table 5 are not clear. The text have to explain how they are calculated

Reviewer 4 Report

Overall:

The manuscript details a very important and emerging issue relating to the spread of inaccurate information via digital/social media. Yet in its current form, the manuscript requires further coverage of pertinent research/issues relating to the impact of social media/digital media messages on the population and a number of areas for inclusion are suggested. Moreover, further clarity is required into the methodological processes/decisions and approaches within the results section prior to publication.

Abstract. Consider embedding in brackets what fake news is for clarity and replacing ‘contribution’ with ‘study’. Moreover, the second last sentence is very long-winded and should be confined to be more succinct. The results sentence is currently now clear.

Line 27. It would be good to define fake news somewhere in this section early. Also, need to ensure consistency throughout the manuscript e.g. describe as inaccurate or not validated later on).

Page 2. There is now research published which can use media information/social media responses to gain feedback on potential public health policy proposals. For instance, see research on proposed heat protection policy for Australian schools in the Health Promotion Journal of Australia.

Page 2. There is growing research of the problems around social media messages/posts. Consider embedding further social media concerns from the research here around privacy, posts ending up with unintended users, concerns how to use the platform, who to follow and how people portray themselves in an inauthentic manner. It would be valuable to consider research related to the authenticity of social media posts and identities.

Section 1.2. It would be useful to have some strong content on globalisation influences within this section and how information intended for a specific context or users, could be received in an entirely different or unsuitable global context with that information.  

Section 1.2. There is recent research relating to how global digital media reporting can influence the discourses within professions and public portrayals of specific professions. Research has recently been released into the role of mass media and the messages and dominant discourses that are communicated to the public is an area that is still under researched and requires further interrogation. For instance, the physical education teaching profession is often portrayed in the digital media over a number of years relating to unhelpful physical crisis messages (see Sport, Education and Society journal) or bullying-type discourses in movie scenes.

The authors should also consider work relating to ‘public pedagogies’ in which young people are constructing their learning/knowledge from digital media immersion beyond classroom learning (Wrench & Garrett, 2018). Others like Ulmer (2016) argue that mass digital media provides the public an ‘entry point’ into the debates. It would be valuable to have further insights into the digital media and how points of view are generated which strengthen or undermine support for specific policies, practices, and ideologies. Others describe media influence as being a ‘fourth estate’ (power of influence on social life subjects).

Page 3. Research showing the “reach of Twitter codes” in how information on Twitter can be spread would be terrific in these sections. Some example authors include Pill et al examining “novel research approaches” which gauged a rolling Twitter conversation and Greenhalgh/Kohler titled “28 days later”.  This type of research mentioned in the above sections are important if the focus is on social media impact.

Page 4, Section 2.1, Paragraph 1. Not clear why just these social media platforms have been selected in comparison to others which have millions of users?

Page 4. It is unclear why these topics were focused upon and why nutrition has been a focus with such little data available? More insight would be valuable to justify why the specific social media platforms and topics were chosen instead of other social media platforms and topics. Currently, the different topics/users/platforms/data collected/information sources make it difficult for comparison. For example, some topics are not available via some of the social media platforms selected.

Page 5. More detail would be valuable relating to the qualitative content analysis procedure, key steps/components of qualitative research addressed in the study (e.g. trustworthiness etc), how themes were generated,

Page 5, Table 3. Please align this table better for clarity with the codes mentioned. Further clarity of the codes is required for the audience to make sense of the results throughout.

Page 6, 2.3. The ethical approval considerations should really be listed earlier and the criteria should be clearly specified. Reference to university human research ethics approval also important.

Page 7, 3.1 title. Suggest deletion of “are Mostly Aggressive”, as the trends associated with the findings should be discovered by the audience through objective articulation within the paragraph sections- not the title.

Lines 277-278. This is a big assumption/statement which requires citation support.

Line 310. Grammar here. Please replace “done”.

Section 3.3. Why weren’t other popular platforms used? E.g. LinkedIN, Instagram, Youtube and so forth?

Section 4. Good opportunity to refer to how global spread of digital information can impact on the discourses associated with a profession. There is growing research into this area.

Line 396. More information would be valuable into the methodology with reduced length of the results would help justify these conclusive statements.

Section 5. Very big paragraph. Please review where the text can be broken into smaller paragraphs.

Line 407. Word choice of “arena”.

Section 4-5. A clear section with specific, structured recommendations from the research would be highly valuable for the audience.

Overall. Further clarity is also needed for readers relating to whether the “aggressive messages” on social media are via users’ specific posts or the content embedded within media articles shared. Please also double-check the sizing of some paragraphs throughout the manuscript.

Round 2

Reviewer 4 Report

The authors should be applauded for careful alignment of the manuscript according to the requests of the reviewer/s. The publication now more additionally covers useful and insightful background information relating to social media impact/influences. One final aspect to consider for the authors is the use of SISM in the title. Social media is relatively new in the research world and therefore any acronyms would not be well established for the wider global audience/international research community. The title also does not provide enough opportunity to spell out this acronym for many readers of this journal (especially from other disciplines) which would not be familiar with this wording. Consideration of modifying the title for clarity (with the acronym still applied throughout the manuscript) could be highly worthwhile. 

Author Response

Thank you very much for your comment.  Considering your last recommendation, we have modified our title.

Now the title is:

A New Application of Social Impact in Social Media for Overcoming Fake News in Health